# Prognostic Factors in Neurorehabilitation of Stroke: A Comparison among Regression, Neural Network, and Cluster Analyses

**DOI:** 10.3390/brainsci11091147

**Published:** 2021-08-29

**Authors:** Marco Iosa, Giovanni Morone, Gabriella Antonucci, Stefano Paolucci

**Affiliations:** 1Department of Psychology, Sapienza University of Rome, 00185 Roma, Italy; gabriella.antonucci@uniroma1.it; 2IRCCS Fondazione Santa Lucia, 00179 Roma, Italy; g.morone@hsantalucia.it (G.M.); s.paolucci@hsantalucia.it (S.P.)

**Keywords:** rehabilitation, cerebrovascular accident, psychometry, accuracy, sensitivity, specificity

## Abstract

There is a large body of literature reporting the prognostic factors for a positive outcome of neurorehabilitation performed in the subacute phase of stroke. Despite the recent development of algorithms based on neural networks or cluster analysis for the identification of these prognostic factors, the literature lacks a rigorous comparison among classical regression, neural network, and cluster analysis. Moreover, the three methods have rarely been tested on a sample independent from that in which prognostic factors have been identified. This study aims at providing this comparison on a wide sample of data (1522 patients) and testing the results on an independent sample (1000 patients) using 30 variables. The accuracy was similar among regression, neural network, and cluster analyses on the analyzed sample (76.6%, 74%, and 76.1%, respectively), but on the test sample, the accuracy of neural network decreased (70.1%). The three models agreed in identifying older age, severe impairment, unilateral spatial neglect, and total anterior circulation infarcts as important prognostic factors. The binary regression analysis also provided solid results in the test sample, especially in terms of specificity (81.8%). Cluster analysis also showed a high sensitivity in the test sample (82.6%) and allowed a meaningful easy-to-use classification tree to be obtained.

## 1. Introduction

There is a wide body of literature reporting the prognostic factors related to an effective neurorehabilitation in patients with stroke in the subacute phase. These are helpful for predicting outcomes and are a fundamental aspect for healthcare resource allocation in order to adequately inform patients and family members and to plan the post-hospital discharge phase [1,2]. A severe neurological impairment at admission, basal disability, older age, aphasia, and unilateral spatial neglect (USN) resulted in prognostic factors of a reduced recovery in terms of mobility [3]. Depression [4] and low active participation in rehabilitation sessions [5,6] were also two negative factors affecting the outcomes of neurorehabilitation. Most of the above studies used logistic regressions to identify these factors, but none tested them in a predictive manner on another independent sample. 

New algorithms are now available and are quoted as potentially more effective for predicting neurorehabilitation outcomes. At the end of the 1990s, two pioneering studies had already suggested the use of machine learning algorithms to identify the prognostic factors of neurorehabilitation outcomes [7] and to predict the following changes in the subacute phase [8]. The recent development of artificial intelligence (AI) is facilitating the diffusion of machine learning in further studies [9,10,11,12]. The prognostic factors identified by AI, usually with an accuracy ≥ 70%, were similar to those classically accounted for: clinical test scores at admission, time from stroke onset to rehabilitation admission, age, sex, body mass index, and dysphasia [12]. Some other studies successfully used neural networks for assessing specific outcomes such as independency in toileting [13] or return to work [14,15].

A third possible approach for the identification of prognostic factors is the use of cluster analysis to draw a classification tree of the outcome [16,17,18]. In fact, cluster analysis is helpful for classifying patients, and different methods can be used with this purpose. The identification of clusters has often been associated with a tree in which each branch is related to a specific condition, but these trees have rarely been tested on wide samples independent from those on which they were computed. Again, older age, severe impairment at admission, and presence of USN have been identified as negative prognostic factors in the cluster of patients with a low recovery [18].

Few sporadic studies have aimed at comparing two of these three methods (such as logistic regression vs. cluster analysis [17] or logistic regression vs. neural network [14]) or further testing their predictive accuracy on a sample independent of the originally analyzed dataset [16].

The aim of this study is to compare the accuracy of the above three approaches (linear regression, neural network, cluster analysis), identifying the prognostic factors of a positive neurorehabilitation outcome in a wide group of patients with stroke, and to verify the results in an independent wide sample of other patients with stroke.

## 2. Materials and Methods

This study was a secondary analysis conducted on a large database used in different previous studies [3,4,19] and further augmented with new data. Because our hospital is also an institute of research, at admission, patients signed an informed consent for the utilization of their data for translational research. The data collection referred to a wide sample of 2522 patients enrolled along a wide period from 1990 to July 2021. The entire sample was randomly divided into a proportion of 3:2 in a sample analyzed by the three models for identifying the prognostic factor (1522 patients) and in a test sample for evaluating the prediction accuracy of the three models (1000 patients). For each patient, the data collected referred to demographical and clinical factors (including risk factors) assessed at admission to a neurorehabilitation hospital. More than 30 variables were assessed; 30 of them were collected for all the patients and used in this study. The used variables are those reported in Table 1. All variables were recorded as dichotomous depending on the presence or absence of the event (for details about how they were assessed, see previous studies [3,4,19]). At discharge, the Barthel Index (BI) was assessed and considered the main outcome of neurorehabilitation; this is a clinical scale evaluating the functional independence of a patient in performing activities of daily living. A positive outcome was associated with a BI-score at discharge > 75 (100 being the maximum achievable BI-score). Conversely, a negative outcome was considered a BI-score at discharge ≤ 75, emergency transfer to another hospital without returning to neurorehabilitation, or death. No binary variables were dichotomized, as reported in Table 1. 

Regression analysis was conducted using a forward binary logistic regression performed to identify, among the 30 analyzed factors, those significantly associated with a positive outcome. Values of coefficient Beta were computed as well as their relevant standard error (SE), their exponential values coinciding with the odds ratio (OR), the relevant *p*-value (statistically significant if <0.05), and the 95% confidence interval (95%CI). 

The artificial neural network analysis was conducted by the ARIANNA model (ARtificial Intelligent Assistant for Neural Network Analysis), already used in other studies [14,15]. It is based on a multilayer perceptron procedure, and it is formed by the input layer (in which the above listed 30 variables are entered), two hidden layers (5 elements in each one), and a final output layer (the output of which was the predicted outcome). The architecture of the ARIANNA was that of a feed forward neural network (FFNN), with data moving in only one direction, from the input nodes through the two hidden layers to the output node. The activation function for all the units in the hidden layers and for the output layer was a hyperbolic tangent. The chosen computational procedure was based on an online training [14,15].

Cluster analysis was based on an algorithm using the growing method of chi-squared automatic interaction detector with a maximum 3 possible levels of nodes starting from the main one. The probabilities of a positive outcome were computed for each one of the identified samples on the analyzed sample. The main output of the cluster analysis was a classification tree. A probability > 50% of obtaining on a BI-score > 75 at discharge was associated with a possible positive outcome tested on the second independent sample following the classification tree.

To compare the three methods, we assessed the accuracy (the overall percentage of cases correctly classified), the sensitivity (the percentage of true positives on all the patients with a good outcome), and the specificity (the percentage of true negatives on all the patients with a bad outcome). All the analyses were performed using IBM SPSS Neural Networks module of IBM SPSS Statistics, version 23 (IBM Corp., Armonk, NY, USA).

## 3. Results

The binary logistic regression showed an accuracy of 76.6% and identified eight statistically significant prognostic factors, as shown in Table 2. The accuracy was influenced by a high specificity (82.9%) and a moderate sensitivity (66.9%), as shown in Table 3.

The neural network analysis showed an accuracy of 74.0%, with a specificity of 80% and a sensitivity of 64.1%. The FFNN associated a weight of importance with each one of the assessed variables. To allow a comparison with the eight factors identified by regression analysis, the first eight prognostic factors in terms of normalized percentage importance of FFNN were: global aphasia (100%), older age (79%), USN (75.4%), low BI-score (69.4%), total anterior circulation infarct according to Bamford classification (TACI, 52.3%), epilepsy (43.6%), depression pre-stroke (38.7%), and thalamic hemorrhage (37.6%). As shown in Table 1, the two methods agreed in the identification of the main five factors, despite assigning a different importance to each one of them. 

Cluster analysis allowed for identifying the classification tree reported in Figure 1. Five factors have been identified as playing a significant role, again all already entered into the regression model. A positive outcome was more probable if at admission, the BI-score was >20 and the patient did not show deficits related to unilateral spatial neglect. Conversely, negative prognostic factors were older age, TACI, and time from stroke to the beginning of neurorehabilitation longer than 1 month from the acute event. All the factors identified by cluster analysis were also entered into the regression model. The accuracy of cluster analysis was 76.1%, with a specificity of 73.7% and a sensitivity of 80%.

When applied to the other dataset of 1000 patients, the binary logistic regression even increased its accuracy (78.5%), thanks to an increment in the sensitivity (72.5%), whereas specificity slightly decreased (81.8%). A similar trend was observed for cluster analysis with an accuracy of 78.2%, a sensitivity of 82.6%, and a specificity of 75.8%. The FFNN lost in sensitivity (58.3%), maintaining a moderate level of specificity (77.6%), with an accuracy of 70.7%, as reported in Table 3.

## 4. Discussion

The main result of our study is a consistent overlapping among regression, neural network, and cluster analyses both in terms of accuracy (sensitivity and specificity) and in terms of identified factors. The simplest method seemed to be the cluster analysis that provided an easy-to-use classification tree. The binary logistic regression had the advantage of identifying some prognostic factors, associating each of them with an odds ratio. Another important result of this comparison was that the cluster analysis was more sensitive, whereas regression analysis was more specific. The neural network lost in accuracy (and, in particular, in sensitivity), passing from the analyzed to the test sample. This could be mainly due to the use of dichotomized variables. In fact, other studies showed a good prognostic ability of machine learning algorithms with ordinal and continuous (not binary) variables [14,15]. Probably the simplest output was given by the cluster analysis, in which five factors allowed for accurately predicting a positive outcome, and two factors were sufficient for a general prognosis.

The factors identified as having a prognostic value were similar between the three methods: older age, TACI, USN, and severe impairment (low BI-score at admission) were factors entered into the model of logistic regression, with an importance higher than 40% for the neural network, and the key factors of the classification tree of cluster analysis. 

Older age, severity of impairment, and USN have already been identified as negative prognostic factors [3]. The role of USN on functional outcome was previously highlighted, suggesting the need of specific training for reducing USN and, in turn, functional deficits [20]. Another prognostic factor was a stroke classified as TACI in terms of Bamford’s classification. Obviously, total anterior circulation infarcts could be more disabling than infarctions in more limited areas. An early beginning of neurorehabilitation (time from stroke less than 1 month) was a prognostic factor for the regression analysis, a key factor of cluster analysis, and it had an importance of 32% in the neural network, in line with the literature [21]. Global aphasia, also reported as a negative prognostic factor in the literature [3], was a factor entered into the model of regression analysis and the most important for FFNN. The role played by USN and aphasia in the recovery of independence in daily living activities suggested, once again, an intertwined connection between motor and cognitive functions [22]. Interestingly, global aphasia remained outside the classification tree of cluster analysis. According to the percentages of sensitivity and specificity, it is conceivable that global aphasia could be important to correctly identify a negative outcome (specificity), but less to identify a positive outcome (sensitivity).

Some factors determined to be prognostic in previous studies such as ischemic stroke [19], depression [4], or gender [23] were not identified as statistically significant in the present study by any of the tested methods. This could be due to many methodological reasons that are outside the focus of our study aiming at comparing the accuracy of the three methods. 

Family support and, surprisingly, smoking before stroke were also entered into the model of binary regression analysis as prognostic factors for a positive outcome. It is well known that informal caregivers, such as familiars, are often involved in the assistance of patients after their return home [24], but our study also showed the importance of their frequent visits during neurorehabilitation. The psychological support of familiars during neurorehabilitation seems to play an important role, as demonstrated for patients with acquired brain injury, with caregivers’ psychological well-being associated with the functional recovery of their loved ones [25]. The role of smoking is controversial: most of the studies agreed that it is a risk factor for having a stroke, and some studies reported that smoking is also a negative factor for rehabilitation [26,27]; however, some others reported that the consumption of nicotine could be a positive prognostic factor for neurorehabilitation [28,29]. Some authors highlighted the importance of adjusting the analyses for many covariates, considering possible baseline differences between smokers and non-smokers, obtaining a null effect of smoking, neither positive nor negative [30]. Further studies should consider the quantitative amount of nicotine consumed before stroke to deeply investigate its potential role on neurorehabilitation outcome. 

FFNN also highlighted the role of smoking (35.6%), but more for epilepsy (43.6%). Epilepsy was another negative prognostic factor of functional recovery, as demonstrated, for example, in patients with subarachnoid hemorrhage [31]. 

There are other prognostic factors reported in the literature, such as body mass index (with the so-called obesity paradox) [32], thrombolysis in acute phase [33], incontinency [34], inflammatory biomarkers [35], or personal USN [36], but they were not assessed for all the patients recorded in the database used in this study, and we did not count them for the comparison among the three methods. However, the main aim of this study was to compare the performances of the three analyses. 

The main limit of this study was probably the dichotomization of continuous variables (such as age or number of smoked cigarettes) and ordinal variables (such as the BI-score) that could have mainly affected the FFNN. Another limit is the selection of the tested variables: demographic (such as schooling level) or clinical factors (such as the assessment of other cognitive functions: for example, attention, memory, and orientation) were not considered in this study, but they should be taken into account in further studies. Finally, another possible limitation is that all the data were collected in a single Italian hospital, and there is the possibility that the results could have been a bit different in other countries, with different health systems, or even in different hospitals, with different settings.

## 5. Conclusions

The three models showed similar accuracies on the analyzed sample and agreed in the identification of the most important factors already recognized by literature as limiting the possibilities of a positive outcome: older age, severe impairment, presence of USN, TACI stroke, aphasia, and time from stroke acute event and the beginning of neurorehabilitation. 

Binary logistic regression was confirmed as a solid approach for binary variables, even more than a complex neural network. Binary logistic regression was helpful in identifying the most important prognostic factors, whereas FFNN provided a continuous assessment of the effects of all the considered factors and probably could be better used for continuous input variables. Binary logistic regression showed, in particular, a good level of specificity, whereas cluster analysis was the most sensitive approach for identifying a positive outcome of neurorehabilitation for patients with stroke.

## Figures and Tables

**Figure 1 brainsci-11-01147-f001:**
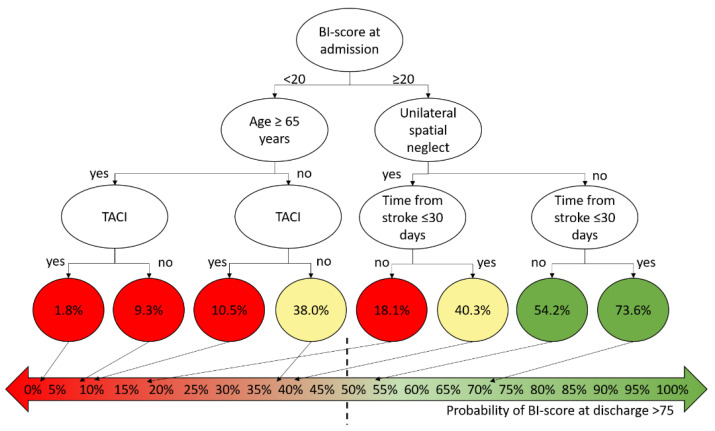
The classification tree obtained by cluster analysis on the analyzed sample.

**Table 1 brainsci-11-01147-t001:** The collected variables expressed as mean ± standard deviation or percentage relative frequency (N: number of patients, BI: Barthel Index, PACI: partial anterior circulation infarcts, TACI: total anterior circulation infarcts, LACI: lacunar infarcts, POCI: posterior circulation infarcts, MCA: middle cerebral artery territory).

Variables		Total Sample	Analyzed Sample	Test Sample
**General** **Variables at Admission**	N	2522	1522	1000
	Age (years)	67.9 ± 13.7	68.0 ± 13.9	67.9 ± 13.4
	Time from stroke (days)	31.7 ± 25.8	31.8 ± 26.5	31.4 ± 24.6
	BI-score	30.1 ± 27.3	30.9 ± 27.3	28.9 ± 27.4
**Analyzed Dichotomous Factors**	Gender (male)	53.6%	53.8%	53.2%
	Age ≥ 65 years	66.0%	66.7%	64.8%
	Time from stroke ≤ 30 days	63.4%	63.5%	63.2%
	BI-score at admission < 20	45.5%	43.9%	48.0%
	Side of stroke (Right)	55.6%	54.3%	57.6%
	Type of stroke (Ischemic)	83.9%	84.4%	83.1%
	Family support (≥3 visits/week)	83.7%	85.0%	81.7%
	Hypertension	60.0%	60.1%	60.0%
	Heart problems	33.5%	34.2%	32.6%
	Diabetes	18.6%	18.8%	18.2%
	Smoker	16.3%	16.8%	15.5%
	Other risk factors	42.2%	43.6%	40.0%
	Depression post-stroke	33.7%	34.0%	33.4%
	Depression pre-stroke	1.5%	1.7%	1.1%
	Epilepsy	6.8%	6.9%	6.7%
	Bamford classification PACI	44.4%	44.4%	44.3%
	Bamford classification TACI	18.0%	18.0%	18.1%
	Bamford classification LACI	11.4%	11.8%	10.8%
	Bamford classification POCI	9.9%	10.2%	9.5%
	Infarctions in MCA	54.3%	53.1%	56.3%
	Lacunar infarctions	11.1%	11.6%	10.5%
	Uncertain territories	7.4%	8.1%	6.3%
	Vertebrobasilar infarctions	9.9%	10.1%	9.6%
	Putaminal hemorrhages	4.9%	4.6%	5.3%
	Thalamic hemorrhages	3.9%	3.5%	4.5%
	Lobar hemorrhages	7.3%	7.4%	7.1%
	Broca’s aphasia	14.9%	15.6%	13.8%
	Wernicke’s aphasia	3.6%	3.1%	4.3%
	Global aphasia	15.4%	14.0%	17.5%
	Unilateral Spatial Neglect	21.1%	21.9%	20.0%
**Discharge Outcomes**	Discharged at home	87.7%	88.8%	86.0%
	Deaths	2.5%	2.7%	2.1%
	Transferred in emergency	9.8%	8.5%	11.9%
	BI-score at discharge	63.6 ± 30.8	64.3 ± 30.5	62.5 ± 31.2
**Analyzed Outcome**	BI-score > 75	37.7%	39.1%	35.6%

**Table 2 brainsci-11-01147-t002:** Variables entered into the model of binary linear regression (Beta: coefficients, SE: standard error, *p*: probability value for rejecting null hypothesis, OR: odds ratio, 95%CI: confidence interval at 95%), their relevant normalized percentage importance in feed forward neural network (FFNN), and the level at which they compare to the classification tree of cluster analysis.

Factors	Binary Logistic Regression					FFNN Importance	Cluster Level
	Beta	SE	*p*	OR	95% CI		
Low BI-score	−2.225	0.159	<0.001	0.108	0.079–0.148	Instrumented assessment of visuomotor coordination in patients with stroke after neurorehabilitation	1°
Neglect	−1.599	0.196	<0.001	0.202	0.138–0.297	75.4%	2°
Global aphasia	−1.422	0.289	<0.001	0.241	0.137–0.425	100%	Excluded
Older Age	−0.971	0.143	<0.001	0.379	0.286–0.501	79.0%	2°
TACI	−0.545	0.270	0.043	0.580	0.342–0.984	52.3%	3°
Time from stroke	0.812	0.148	<0.001	2.251	1.685–3.007	32.0%	3°
Family support	0.453	0.194	0.019	1.573	1.076–2.299	16.7%	Excluded
Smoker	0.394	0.184	0.032	1.484	1.034–2.128	35.6%	Excluded

**Table 3 brainsci-11-01147-t003:** Accuracy, sensitivity, and specificity percentages of the three applied analyses for the analyzed and tested samples.

Sample	Parameters	Regression Analysis	Neural Network	Cluster Analysis
Analyzed sample (*N* = 1522)	Accuracy	76.6%	74.0%	76.1%
	Sensitivity	66.9%	64.1%	80.0%
	Specificity	82.9%	80.0%	73.7%
Test Sample (*N* = 1000)	Accuracy	78.5%	70.1%	78.2%
	Sensitivity	72.5%	58.3%	82.6%
	Specificity	81.8%	77.6%	75.8%

## Data Availability

The complete analysis outputs of this study are available to anyone requiring them upon request to the corresponding author.

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
