# Peer review of "Prognostic Factors in Neurorehabilitation of Stroke: A Comparison among Regression, Neural Network, and Cluster Analyses"

_brainsci, 2021, doi:10.3390/brainsci11091147_

Round 1

Reviewer 1 Report

This study takes profit of the data obtained from a relatively large sample of stroke patients, and allows to examine the relative accuracy of two different classical statistical methods, as well as a method based on machine learning, to determine the prognostic value, for the sake of neurorehabilitation, of a set of different variables. The fact that the models were also tested, in a predictive manner, in an independent sample is particularly valuable.

Although I agree with the limitations indicated by the authors, I would add another limitation that should be taken into account: the existence of possible regional differences with regard to the prognostic value of the factors that were analyzed. This study was carried out in Italy, and the possibility that the results could have been a bit different in other countries, with different health systems, cannot be disregarded.

In addition, there are also some minor corrections required: 1) Be careful to introduce all the abbreviatures the first time they are used; 2) Although English editing seems rather careful, there are nevertheless a few odd sentences. For example, the last paragraph of discussion); 3)There are errors in the numbers of the references, and subsequent citations in the text. 

Author Response

AUTHORS: We would like to thank the Editor and the Reviewers for their really positive judgments about our work and for the qualified suggestions that helped us into improving this manuscript in this revised version. In the following we provided a detailed point-by-point responses to the few comments of the reviewers, also reporting between apices the rephrased sentences as required.

REVIEWER 1

This study takes profit of the data obtained from a relatively large sample of stroke patients, and allows to examine the relative accuracy of two different classical statistical methods, as well as a method based on machine learning, to determine the prognostic value, for the sake of neurorehabilitation, of a set of different variables. The fact that the models were also tested, in a predictive manner, in an independent sample is particularly valuable.

AUTHORS: We would like to thank the Reviewer for the positive judgment about our work.

Although I agree with the limitations indicated by the authors, I would add another limitation that should be taken into account: the existence of possible regional differences with regard to the prognostic value of the factors that were analyzed. This study was carried out in Italy, and the possibility that the results could have been a bit different in other countries, with different health systems, cannot be disregarded.

AUTHORS: Thank you very much for this suggestion, we agree with the Reviewer that it could be a possible limit of our study. We have now added the following sentence into the revised version of the manuscript:
“Finally, another possible limitation is that all the data were collected in a single Italian hospital, and there is the possibility that the results could have been a bit different in other Countries, with different health systems, or even in different hospitals, with different settings.”

In addition, there are also some minor corrections required: 1) Be careful to introduce all the abbreviatures the first time they are used; 2) Although English editing seems rather careful, there are nevertheless a few odd sentences. For example, the last paragraph of discussion); 3)There are errors in the numbers of the references, and subsequent citations in the text. 

AUTHORS: We have corrected the parts in which abbreviations were introduced without a clear explication. We have done a carefully English editing and a check about typos. Thanks for having noted the error about reference numeration, in the bibliography the number 1 was repeated twice, affecting the numeration of all the other references: we have now corrected it. About the last paragraph of the discussion we have rephrased it as follows:

The main limit of this study was probably the dichotomization of continuous variables (such as age, or number of smoked cigarettes) and ordinal variables (such as the BI-score) that could have mainly affected the FFNN. Another limit is the selection of the tested variables: demographical (such as schooling level) or clinical factors (such as the assessment of other cognitive functions, for example attention, memory and orientation) were not considered in this study, but they should be taken into account in further studies.

REVIEWER 2

The work is well written and although it essentially focused on evaluating statistical approaches, it also provides clinicians with valuable support in assessing risk factors associated with stroke recovery.

This is why it also allows us to identify a different weight between factors known to be associated with recovery.

for what is defined to be the purpose of the study, the methods and produced references appear to be sufficient and adequate and except for a minimal revision of English to be entrusted to a native speaker.

AUTHORS: We would like to thank the Reviewer for the positive judgment about our work. In this revised version of our manuscript we have carefully performed an English editing correcting for typos and odd sentences.

Reviewer 2 Report

the work is well written and although it essentially focused on evaluating statistical approaches, it also provides clinicians with valuable support in assessing risk factors associated with stroke recovery.

This is why it also allows us to identify a different weight between factors known to be associated with recovery.

for what is defined to be the purpose of the study, the methods and produced references appear to be sufficient and adequate and except for a minimal revision of English to be entrusted to a native speaker.

Author Response

AUTHORS: We would like to thank the Editor and the Reviewers for their really positive judgments about our work and for the qualified suggestions that helped us into improving this manuscript in this revised version. In the following we provided a detailed point-by-point responses to the few comments of the reviewers, also reporting between apices the rephrased sentences as required.

REVIEWER 2

The work is well written and although it essentially focused on evaluating statistical approaches, it also provides clinicians with valuable support in assessing risk factors associated with stroke recovery.

This is why it also allows us to identify a different weight between factors known to be associated with recovery.

for what is defined to be the purpose of the study, the methods and produced references appear to be sufficient and adequate and except for a minimal revision of English to be entrusted to a native speaker.

AUTHORS: We would like to thank the Reviewer for the positive judgment about our work. In this revised version of our manuscript we have carefully performed an English editing correcting for typos and odd sentences.